# Relationship between HIV viral suppression and multidrug resistant tuberculosis treatment outcomes

Keri Geiger[1,2]*, Amita Patil[2], Chakra Budhathoki[1], Kelly E. Dooley[3], Kelly Lowensen[1,2], Norbert Ndjeka[4], Jacqueline Ngozo[5], Jason E. Farley[1,2]

1 School of Nursing, Johns Hopkins University, Baltimore, Maryland, United States of America, 2 Center for Infectious Disease and Nursing Innovation, Johns Hopkins University, Baltimore, Maryland, United States of America, 3 Division of Infectious Diseases, Vanderbilt University Medical Center, Nashville, Tennessee, United States of America, 4 National Department of Health, Tuberculosis Control and Management, Pretoria, Gauteng, South Africa, 5 KwaZulu-Natal Department of Health, Tuberculosis Programme, Pietermaritzburg, KwaZulu-Natal, South Africa

* kgeiger3@jhu.edu

**Data Availability Statement:** All data can be found in the manuscript and supporting information files.

## Abstract

The impact of HIV viral suppression on multidrug resistant tuberculosis (MDR-TB) treatment outcomes among people with HIV (PWH) has not been clearly established. Using secondary data from a cluster-randomized clinical trial among people with MDR-TB in South Africa, we examined the effects of HIV viral suppression at MDR-TB treatment initiation and throughout treatment on MDR-TB outcomes among PWH using multinomial regression. This analysis included 1479 PWH. Viral suppression (457, 30.9%), detectable viral load (524, 35.4%), or unknown viral load (498, 33.7%) at MDR-TB treatment initiation were almost evenly distributed. Having a detectable HIV viral load at MDR-TB treatment initiation significantly increased risk of death compared to those virally suppressed (relative risk ratio [RRR] 2.12, 95% CI 1.11–4.07). Among 673 (45.5%) PWH with a known viral load at MDR-TB outcome, 194 (28.8%) maintained suppression, 267 (39.7%) became suppressed, 94 (14.0%) became detectable, and 118 (17.5%) were never suppressed. Those who became detectable (RRR 11.50, 95% CI 1.98–66.65) or were never suppressed (RRR 9.28, 95% CI 1.53–56.61) were at significantly increased risk of death (RRR 6.37, 95% CI 1.58–25.70), treatment failure (RRR 4.54, 95% CI 1.35–15.24), and loss to follow-up (RRR 7.00, 95% CI 2.83–17.31; RRR 2.97, 95% CI 1.02–8.61) compared to those who maintained viral suppression. Lack of viral suppression at MDR-TB treatment initiation and failure to achieve or maintain viral suppression during MDR-TB treatment drives differences in MDR-TB outcomes. Early intervention to support access and adherence to antiretroviral therapy among PWH should be prioritized to improve MDR-TB treatment outcomes.

## Introduction

Co-infection with tuberculosis (TB) and human immunodeficiency virus (HIV) is common, and South Africa has a particularly high burden of both diseases [1]. Patients with TB resistant to the first-line drugs isoniazid and rifampicin (multidrug resistant or MDR-TB) typically

**Funding:** This work was supported by the National Institute of Allergy and Infectious Disease [NIAID R01AI104488-02; ClinicalTrials.gov Identifier NCT02129244; PI Jason Farley]. KG received funding from the National Center for Advancing Translational Sciences grant number TL1 TR003100. KED is supported by NIAID K24AI150349. The funders had no role in study design, data collection and analysis, decision to publish, or preparation of the manuscript.

**Competing interests:** The authors have declared that no competing interests exist.

have worse treatment outcomes than patients with drug-sensitive TB [1]. HIV co-infection further complicates MDR-TB treatment due to the need to treat both infections simultaneously leading to potential drug-drug interactions, increased pill burden, greater potential for treatment-related adverse effects of medications, and additional complexity [2–7]. However, it remains unclear whether successful HIV treatment including achieving and maintaining viral suppression impacts MDR-TB treatment outcomes among people with HIV (PWH).

In settings with high rates of MDR-TB/HIV co-infection, thoroughly understanding how HIV affects MDR-TB outcomes is a priority. A meta-analysis has shown that HIV co-infection did not significantly decrease the odds of overall MDR-TB treatment success, though PWH were more likely to die during MDR-TB treatment [8]. In other studies conducted across a range of settings including Eastern Europe, sub-Saharan Africa, and Brazil, differences in MDR-TB outcome by HIV status have been demonstrated [9–15]. These differences have been attributed to immune compromise evidenced by low CD4 count and lack of adequate treatment for HIV when PWH were not taking antiretroviral therapy (ART) [9, 12, 14–16].

Though ART use is a prerequisite to achieving HIV viral suppression, taking ART alone is insufficient, as poor treatment adherence and viral resistance to ART can lead to active viral replication indicated by a detectable HIV viral load. It is possible that ongoing HIV replication and its effects on inflammation and cell-mediated immunity could affect MDR-TB treatment outcomes, though this has rarely been explored. While HIV co-infection rates are widely published in the MDR-TB literature, HIV viral load data, particularly longitudinal viral load results, are rarely reported [17].

We investigated the effects of HIV viral suppression at MDR-TB treatment initiation and throughout MDR-TB treatment on MDR-TB treatment outcomes among a cohort of PWH undergoing programmatic treatment for MDR-TB in South Africa.

## Methods

### Parent study

This analysis used data from 2,545 participants in a cluster-randomized clinical trial (clinicaltrials.gov registration number: NCT02129244) of a nurse case management intervention for people with MDR-TB recruited between 2014 to 2020 from 13 hospitals in KwaZulu-Natal and Eastern Cape, South Africa [18]. Participants were randomized by hospital location to have either standard of care or a dedicated nurse case manager who assisted the treating clinician with patient management, including patient teaching; early identification of adverse reactions and symptoms; adherence counseling; follow-up of laboratory values; and assisting the outreach teams with tracing patients who missed appointments. Standard of care for MDR-TB in South Africa generally involved management by a clinical provider, usually a physician and hospitalization at a TB-specific treatment center for a period of several weeks to months depending on individual clinical condition, followed by monthly follow-up visits for the duration of MDR-TB treatment [19]. No nurse case managers were involved in care in for participants in the intervention arm. For people with MDR-TB and HIV, HIV care including provision of antiretrovirals was usually managed by the clinician treating TB for the duration of MDR-TB treatment [19]. Inclusion criteria for the parent study included being 13 years or older and having microbiologically confirmed rifampicin-resistant or MDR-TB at enrollment. According to South African clinical guidelines, rifampicin resistant TB was treated clinically as MDR-TB [19]. Data from the intervention and the control arm of the parent study were included in this analysis, and it is possible that the nurse case manager intervention could have led to better HIV and MDR-TB outcomes for participants in the intervention arm. Therefore, the presence of the nurse case manager was controlled for statistically.

## Sample

For the evaluation of HIV viral load at MDR-TB treatment initiation on MDR-TB outcome, we included all participants in the parent study with available and clean data at the time of analysis in October 2023 who were known to be HIV positive or newly tested HIV positive at MDR-TB treatment initiation and had an MDR-TB outcome of treatment success, treatment failure, loss to follow-up, or death. We next evaluated HIV viral load status throughout the MDR-TB treatment period. We excluded participants from the prior model if they did not have an available HIV viral load result at the time of MDR-TB treatment outcome, defined as three months prior to through six months after MDR-TB outcome date. HIV viral load was measured outside of the study under program conditions and was captured in the National Health Laboratory Service (NHLS) database. The study sample is described in Fig 1.

## Statistical analyses

We used descriptive statistics to compare demographic and clinical characteristics of PWH at the time of MDR-TB treatment initiation. We classified those with an HIV viral load result available at the time of MDR-TB treatment outcome into four categories based on how HIV viral load results changed during the MDR-TB treatment period. The four categories were: (a) 'maintained suppression' if the HIV viral load was suppressed at both MDR-TB treatment initiation and MDR-TB outcome, (b) 'never suppressed' if HIV viral load was detectable at both MDR-TB treatment initiation and MDR-TB outcome, (c) 'became detectable' if HIV viral load was detectable at MDR-TB outcome and either suppressed or unknown at MDR-TB treatment initiation, and (d) 'became suppressed' if HIV viral load was suppressed at MDR-TB outcome and either detectable or unknown at MDR-TB treatment initiation, as indicated in Fig 1. We report the frequency and proportion of participants who fell into each category by MDR-TB outcome. Due to the infrequency of HIV viral load monitoring during MDR-TB treatment, we are not able to comment on any fluctuations in viral load that may have occurred during the MDR-TB treatment period, as intermediate viral load was not routinely measured or widely available for participants.

We built two separate multinomial regression models to predict MDR-TB treatment outcome, first using HIV viral load at the time of MDR-TB treatment initiation and separately according to the categories of HIV viral suppression described above. For both models, we tested the effects of HIV viral suppression with a bivariate model before adding covariates to control for other factors known to affect MDR-TB outcome. For the first model, covariates included age, sex, baseline CD4 count, and BMI. For the second model, covariates included age, sex, length of MDR-TB treatment, MDR-TB treatment regimen, and arm of the parent study. Results of the multinomial regression models are reported as relative risk ratios describing the risk of a particular MDR-TB treatment outcome compared to the risk of MDR-TB treatment success, the base category, given differences in the predictor variables, according to the standard definition of multinomial regression output [20, 21]. Relative risk ratio is the exponentiated regression coefficient for multinomial regression in Stata version 16, used for all statistical analysis [22]. Both models accounted for cluster randomization in the parent study using cluster-correlated robust estimation of variance [23, 24].

## Variable definitions

The primary outcome in this study was MDR-TB treatment outcome, defined according to the World Health Organization's published definitions in use during the parent study but combining cure and treatment completion into one category called MDR-TB treatment success [25]. The primary predictor was HIV viral suppression, defined as a viral load below the detectable

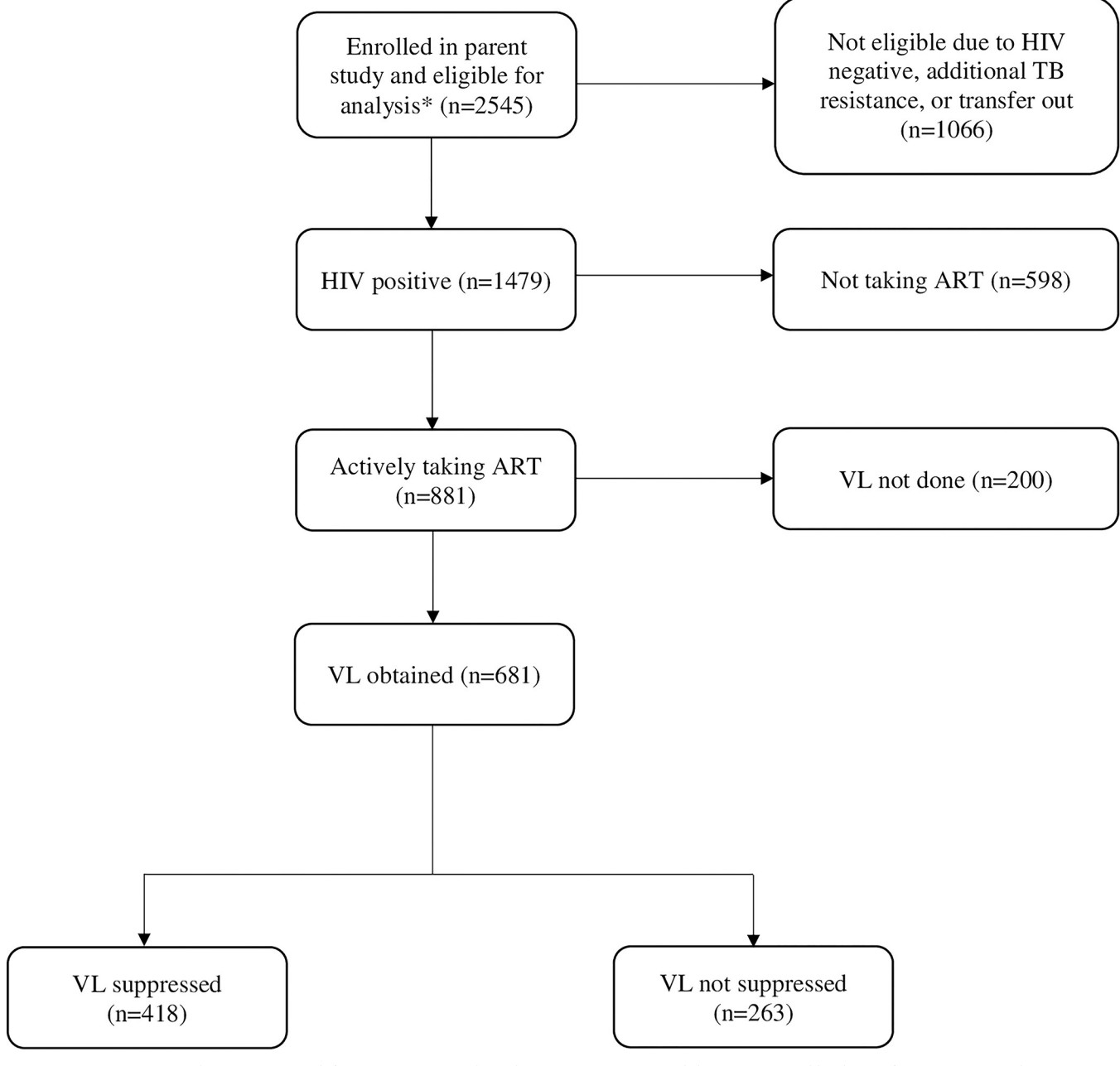

**Fig 1. Study sample.** HIV, human immunodeficiency virus; TB, tuberculosis; ART, antiretroviral therapy; VL, viral load; transfer out, parent study participant was transferred to a different MDR-TB treatment site prior to MDR-TB outcome which did not participate in the parent study. *At the time of this analysis, data was available for 2545 of 2890 participants in the parent study.

limit of the available assay or less than 400 copies per milliliter in accordance with South African national guidelines [26]. HIV viral load results taken within three months prior to or six months after the date of MDR-TB treatment outcome were included in this analysis as outcome results, and HIV viral load results taken within 12 months prior to or four weeks after the first antitubercular medication was taken were included as results at MDR-TB treatment initiation.

Baseline CD4 count was included if tested within 12 months prior to or four weeks after the first MDR-TB medication was taken. In the second model, HIV viral load status was classified in the four categories defined above. MDR-TB treatment length was defined as the number of days between the first day that MDR-TB medications were taken and the date of MDR-TB outcome. MDR-TB treatment regimen was categorized as regimens that used injectable medications, all-oral regimens using the newer antitubercular agent bedaquiline, both injectable and bedaquiline-based all-oral regimens due to the clinician switching regimens during treatment, or individualized regimen if neither bedaquiline nor injectable medications were used. Finally, the variable "arm of the parent study" was defined by treatment site as having a nurse case manager or receiving standard of care.

## Missing data

We searched for all missing HIV viral load and CD4 count results in the NHLS database, which is a repository for all laboratory data from every public health facility in South Africa [27]. Others have used the NHLS as a marker for engagement in care among PWH in South Africa, as it is considered accurate and complete [28, 29]. If, after the search was completed, no HIV viral load results returned, then HIV viral load was considered clinically unavailable; that is, never ordered, never collected, or blocked from analysis following collection as a result of electronic gate keeping restricting the frequency of HIV viral load testing. Participants with an unavailable HIV viral load were coded as unknown viral load for the analysis of treatment initiation viral load and excluded from the analysis of viral load over time if the outcome viral load was unavailable. Because of the large proportion of unavailable HIV viral load results at MDR-TB treatment outcome and the importance of this variable as the outcome of interest, we did not impute HIV viral load results, choosing instead to label the viral load as 'unknown' and limit the sample to those with known results. After completing the NHLS search and entering additional CD4 results, 191 (12.9%) participants did not have an available CD4 count. We used multiple imputation to impute these into the data set. We also used multiple imputation to account for the 3.8% of participants with unknown BMI at MDR-TB treatment initiation. No other covariates had missing values.

## Ethics statement

All participants aged 18 years and older gave formal, written consent to participate in the parent study as a condition of enrollment. Adolescents aged 13 to 17 years gave verbal assent to participate, and a parent guardian, or caregiver signed the formal written consent form. The parent nurse case management study was approved by the Johns Hopkins School of Medicine Institutional Review Board (Application #NA_00078899), the province-level research committees in Eastern Cape and KwaZulu-Natal, and the IRB at the University of KwaZulu-Natal (Application #BE530/14). This sub-study was approved as a change in research protocol to the parent study.

# Results

A total of 1479 parent study participants were eligible for this study [S1 Data]. The mean age was 37.1 years (SD 10.3, IQR 30–43), 809 (54.7%) were male, and median CD4 count was 182 (IQR 75–363). Additional demographic and clinical characteristics of the sample are presented by HIV viral load status MDR-TB outcome in Table 1.

When controlling for age, sex, BMI, and immune function indicated by CD4 count at MDR-TB treatment initiation, the relative risk ratio (RRR) for death was significantly higher than that of MDR-TB treatment success for people with a detectable (RRR 2.12, 95% CI 1.11–4.07) HIV viral load, compared to those who were virally suppressed. Though relative risk

**Table 1. Demographic, disease, and laboratory characteristics of PWH (N = 1479).**

| Total number of participants | | All PWH | Not Taking ART | Taking ART (n = 881) | | |
|---|---|---|---|---|---|---|
| | | | | VL not done^ | VL done (n = 681) | |
| | | | | | VL undetectable^^ | VL detectable^^^ |
| | | **1479 (100%)** | **598 (40.4%)** | **200 (22.7%)** | **418 (61.4%)** | **263 (38.6%)** |
| Age | Mean (SD) | 37.1 (10.3) | 36.2 (9.5) | 36.9 (11.7) | 39.4 (10.7) | 35.9 (9.7) |
| Sex | Male | 809 (54.7%) | 342 (57.2%) | 112 (56.3%) | 205 (49.0%) | 150 (57.0%) |
| | Female | 670 (45.3%) | 256 (42.8%) | 87 (43.7%) | 213 (51.0%) | 113 (43.0%) |
| Number of Prior TB Episodes | None | 649 (43.9%) | 337 (56.4%) | 81 (40.5%) | 159 (38.0%) | 72 (27.4%) |
| | One | 657 (44.4%) | 204 (34.1%) | 86 (43.0%) | 207 (49.5%) | 160 (60.8%) |
| | Two or more | 117 (7.9%) | 35 (5.9%) | 19 (9.5%) | 37 (8.9%) | 26 (9.9%) |
| | Unknown | 56 (3.8%) | 22 (3.7%) | 14 (7.0%) | 15 (3.6%) | 5 (1.9%) |
| Education Level | Less than primary school | 263 (17.8%) | 91 (15.2%) | 31 (15.5%) | 96 (23.0%) | 45 (17.1%) |
| | Primary school complete | 795 (53.8%) | 334 (55.9%) | 111 (55.5%) | 199 (47.6%) | 151 (57.4%) |
| | Beyond primary school | 407 (27.5%) | 168 (28.1%) | 55 (27.5%) | 118 (28.2%) | 66 (25.1%) |
| | Unknown | 14 (1.0%) | 5 (0.8%) | 3 (1.5%) | 5 (1.2%) | 1 (0.4%) |
| Employment Status | Unemployed | 972 (65.7%) | 378 (63.2%) | 134 (67.0%) | 267 (63.9%) | 193 (73.4%) |
| | Employed part-time | 196 (13.3%) | 93 (15.6%) | 26 (13.0%) | 53 (12.7%) | 24 (9.1%) |
| | Employed full-time | 308 (20.8%) | 126 (21.1%) | 40 (20.0%) | 96 (23.0%) | 46 (17.5%) |
| | Unknown | 3 (0.2%) | 1 (0.2%) | 0 (0.0%) | 2 (0.5%) | 0 (0.0%) |
| Housing | Rural or farm | 879 (59.4%) | 301 (50.3%) | 117 (58.8%) | 293 (70.1%) | 167 (63.5%) |
| | Township | 544 (36.8%) | 271 (45.3%) | 75 (37.5%) | 110 (26.3%) | 88 (33.5%) |
| | Urban/CBD or suburban | 52 (3.5%) | 24 (4.0%) | 7 (3.5%) | 14 (3.4%) | 7 (2.6%) |
| | Unknown | 4 (0.3%) | 2 (0.3%) | 0 (0.0%) | 1 (0.2%) | 1 (0.4%) |
| Baseline BMI | Mean (SD) | 20.7 (4.9) | 20.4 (4.7) | 21.1 (5.3) | 21.3 (5.3) | 20.0 (4.2) |
| Baseline CD4 (cells/mm3) | <50 | 240 (16.2%) | 131 (21.9%) | 24 (12.0%) | 13 (3.1%) | 72 (27.4%) |
| | 50–199 | 445 (30.1%) | 216 (36.1%) | 50 (25.0%) | 80 (19.1%) | 99 (37.6%) |
| | 200–499 | 414 (28.0%) | 139 (23.2%) | 44 (22.0%) | 171 (40.9%) | 60 (22.8%) |
| | ≥500 | 194 (13.1%) | 54 (9.0%) | 19 (9.5%) | 110 (26.3%) | 11 (4.2%) |
| | Unknown | 186 (12.6%) | 58 (9.7%) | 63 (31.5%) | 44 (10.5%) | 21 (8.0%) |
| ART Regimen | EFV-based | | | 161 (80.5%) | 342 (81.8%) | 197 (74.9%) |
| | NVP-based | | | 21 (10.5%) | 40 (9.6%) | 16 (6.1%) |
| | LPVr-based | | | 11 (5.5%) | 30 (7.2%) | 41 (15.6%) |
| | Other or unknown | | | 7 (3.5%) | 6 (1.4%) | 9 (3.4%) |

PWH, people with HIV; ART, antiretroviral therapy; TB, tuberculosis; HIV, human immunodeficiency virus; VL, human immunodeficiency viral load; SD, standard deviation; CBD, central business district; BMI, body mass index; mm3, cubic millimeters; ^VL available within 12 months prior to or four weeks after MDR-TB treatment initiation; ^^VL <400 copies/mm3; ^^^VL ≥400 copies/mm3. Note: Some percentages do not add to 100.0% due to rounding.

ratios for treatment failure and becoming lost to follow-up compared to MDR-TB treatment success also increased for both groups relative to those with HIV viral suppression at MDR-TB treatment initiation, the adjusted increased risk did not achieve statistical significance. Table 2 details the adjusted relative risk ratio for negative MDR-TB outcomes compared to treatment success according to baseline HIV viral load.

HIV viral load results at the time of MDR-TB treatment outcome were available for 673 (45.5%) individuals. These participants were further characterized by their MDR-TB treatment initiation and outcome viral load results into the following categories: maintained suppression, became suppressed, became detectable, or never suppressed as described above. HIV viral load category is presented by MDR-TB treatment outcome in Table 3.

**Table 2. Odds ratios for taking ART MDR-TB initiation (n = 1479).**

| | | Univariable Model | | Multivariable Model | |
|---|---|---|---|---|---|
| | | OR | 95% CI | aOR | 95% CI |
| Age (years) | | 1.02** | 1.01–1.03 | 1.01 | 0.99–1.02 |
| Female (ref. male) | | 1.18 | 0.96–1.46 | 1.18 | 0.93–1.51 |
| Prior TB episodes (ref. none) | One | 2.35*** | 1.87–2.96 | 2.58*** | 2.03–3.28 |
| | Two or more | 2.45*** | 1.58–3.78 | 2.82*** | 1.78–4.48 |
| Education level (ref. less than primary school) | Primary school complete | 0.72* | 0.54–0.96 | 0.91 | 0.66–1.27 |
| | More than primary school | 0.75 | 0.54–1.03 | 1.02 | 0.70–1.48 |
| Housing (ref. rural or farm) | Township | 0.52*** | 0.42–0.65 | 0.49*** | 0.38–0.62 |
| | City/CBD or suburban | 0.60 | 0.34–1.06 | 0.52* | 0.29–0.95 |
| CD4 count (cells/mm3, ref. ≥500) | <50 | 0.32*** | 0.22–0.47 | 0.33*** | 0.21–0.50 |
| | 50–199 | 0.40*** | 0.28–0.58 | 0.40*** | 0.27–0.58 |
| | 200–499 | 0.70 | 0.49–1.01 | 0.69 | 0.47–1.01 |
| Employment status (ref. unemployed) | Employed part-time | 0.70* | 0.52–0.96 | 0.65* | 0.47–0.91 |
| | Employed full-time | 0.92 | 0.71–1.20 | 0.96 | 0.72–1.29 |
| Baseline BMI | | 1.02 | 0.99–1.04 | 1.02 | 0.99–1.05 |

HIV, human immunodeficiency virus; PWH, people with HIV; ART, antiretroviral therapy; VL, viral load; MDR-TB, multidrug-resistant tuberculosis; OR, odds ratio; CI, confidence interval; aOR, adjusted odds ratio; ref., reference; TB, tuberculosis; BMI, body mass index; mm3, cubic millimeters

*p<0.05

**p<0.01

***p<0.001

**Table 3. Odds ratios for having an HIV viral load among PWH taking ART MDR-TB initiation (n = 881).**

| | | Univariable Model | | Multivariable Model | |
|---|---|---|---|---|---|
| | | OR | 95% CI | aOR | 95% CI |
| Age (years) | | 1.01 | 0.99–1.03 | 1.01 | 0.99–1.03 |
| Female (ref. male) | | 1.17 | 0.85–1.60 | 1.29 | 0.89–1.85 |
| Prior TB episodes (ref. none) | One | 1.48* | 1.05–2.08 | 1.59* | 1.11–2.28 |
| | Two or more | 1.18 | 0.66–2.11 | 1.24 | 0.67–2.26 |
| Education level (ref. less than primary school) | Primary school complete | 0.73 | 0.47–1.13 | 0.87 | 0.54–1.39 |
| | More than primary school | 0.78 | 0.48–1.27 | 0.97 | 0.57–1.65 |
| Housing (ref. rural or farm) | Township | 0.68* | 0.48–0.95 | 0.68* | 0.48–0.95 |
| | City/CBD or suburban | 0.77 | 0.32–1.86 | 0.73 | 0.30–1.81 |
| CD4 count (cells/mm3, ref. ≥500) | <50 | 0.50* | 0.26–0.96 | 0.48* | 0.24–0.96 |
| | 50–199 | 0.51* | 0.30–0.93 | 0.48* | 0.35–0.90 |
| | 200–499 | 0.72 | 0.41–1.24 | 0.66 | 0.37–1.16 |
| Employment status (ref. unemployed) | Employed part-time | 0.86 | 0.53–1.40 | 0.84 | 0.51–1.38 |
| | Employed full-time | 1.03 | 0.69–1.54 | 1.13 | 0.74–1.74 |
| Baseline BMI | | 0.99 | 0.96–1.02 | 0.98 | 0.94–1.01 |

HIV, human immunodeficiency virus; PWH, people with HIV; ART, antiretroviral therapy; VL, viral load; MDR-TB, multidrug-resistant tuberculosis; OR, odds ratio; CI, confidence interval; aOR, adjusted odds ratio; ref., reference; TB, tuberculosis; BMI, body mass index; mm3, cubic millimeters

*p<0.05

**p<0.01

***p<0.001

**Table 4. Odds ratios for HIV viral suppression among PWH taking ART with a known VL at MDR-TB initiation (n = 681).**

| | | Univariable Model | | Multivariable Model | |
|---|---|---|---|---|---|
| | | OR | 95% CI | aOR | 95% CI |
| Age (years) | | 1.03*** | 1.02–1.05 | 1.04*** | 1.01–1.06 |
| Female (ref. male) | | 1.38* | 1.01–1.88 | 0.94 | 0.62–1.43 |
| Prior TB episodes (ref. none) | One | 0.59** | 0.42–0.83 | 0.64* | 0.42–0.99 |
| | Two or more | 0.63 | 0.36–1.12 | 0.63 | 0.30–1.30 |
| Education level (ref. less than primary school) | Primary school complete | 0.62* | 0.41–0.94 | 1.17 | 0.69–2.01 |
| | More than primary school | 0.84 | 0.53–1.34 | 1.96* | 1.05–3.65 |
| Housing (ref. rural or farm) | Township | 0.71 | 0.51–1.00 | 0.59* | 0.38–0.91 |
| | City/CBD or suburban | 1.15 | 0.46–2.91 | 1.04 | 0.90–2.42 |
| CD4 count (cells/mm3, ref. ≥500) | <50 | 0.02*** | 0.01–0.05 | 0.02*** | 0.01–0.04 |
| | 50–199 | 0.08*** | 0.04–0.15 | 0.06*** | 0.03–0.13 |
| | 200–499 | 0.27*** | 0.14–0.54 | 0.23*** | 0.11–0.46 |
| ART regimen (ref. efavirenz-based) | Nevirapine-based | 1.44 | 0.79–2.64 | 1.66 | 0.84–3.31 |
| | Ritonavir-boosed Lopinavir-based | 0.42** | 0.26–0.70 | 0.38** | 0.19–0.73 |
| | Other or Unknown ART | 0.38 | 0.13–1.09 | 0.42 | 0.11–1.60 |
| Employment status (ref. unemployed) | Employed part-time | 1.59 | 0.95–2.67 | 1.34 | 0.72–2.49 |
| | Employed full-time | 1.51* | 1.01–2.24 | 1.48 | 0.90–2.42 |
| Baseline BMI | | 1.05** | 1.01–1.09 | 1.00 | 0.96–1.05 |

HIV, human immunodeficiency virus; PWH, people with HIV; ART, antiretroviral therapy; VL, viral load; MDR-TB, multidrug-resistant tuberculosis; OR, odds ratio; CI, confidence interval; aOR, adjusted odds ratio; ref., reference; TB, tuberculosis; BMI, body mass index; mm3, cubic millimeters

*p<0.05

**p<0.01

***p<0.001

When controlling for age, sex, length of MDR-TB treatment, MDR-TB regimen, and arm of the parent study, both the 'became detectable' and 'never suppressed' groups had an increased risk ratio for all three negative MDR-TB outcomes relative to MDR-TB treatment success, when compared with those who maintained suppression. Relative risk ratio for MDR-TB treatment outcome for the group who became suppressed during MDR-TB treatment did not differ significantly from that of the group who maintained suppression across any MDR-TB treatment outcome. Results of the multinomial regression analysis detailing relative risk ratio of MDR-TB treatment outcome by HIV viral load category are presented in Table 4.

## Discussion

Our results indicate that detectable HIV viral load at MDR-TB treatment initiation was significantly associated with increased risk of death during MDR-TB treatment. Those whose HIV viral load either became detectable or who never achieved HIV viral suppression during MDR-TB treatment were at significantly higher risk of all poor MDR-TB outcomes including death, treatment failure, and loss to follow-up, relative to MDR-TB treatment success, compared to those who maintained HIV viral suppression. Though several studies have demonstrated differences in MDR-TB outcome by HIV status, few have explored the impact of HIV viral suppression on MDR-TB outcomes [9, 12, 13, 15]. Our results suggest that the differences in MDR-TB outcomes between those with and without HIV co-infection which have been reported by others may actually be due to poorly controlled HIV disease and HIV viral

replication, while the cohorts in which no difference in MDR-TB outcome by HIV status was seen could be due to high rates of viral suppression in these cohorts [8–15]. As HIV viral load is rarely reported, our study is the first to clearly demonstrate this correlation [17].

While this study did not investigate the mechanism underlying the observed association between HIV viral replication and MDR-TB treatment outcome, there are several possible explanations. First, a detectable HIV viral load is often associated with diminished immune function and a low CD4 count. However, our models found a significant association between HIV viral load and MDR-TB treatment outcome even when controlling for CD4 count. Therefore, in this sample, HIV viral replication appeared to influence MDR-TB outcome independent of immune function. A detectable HIV viral load could also be a marker of poor adherence to ART or viral resistance to the ART regimen, leading to worsening HIV disease. Death during MDR-TB treatment was recorded as death from any cause, and as a detectable HIV viral load places people with HIV at higher risk of death due to HIV-related causes, it is possible that some excess death was attributable to HIV alone. If a detectable HIV viral load indicated poor adherence to all medications including those treating MDR-TB, this could also explain the observed association. However, it is unlikely that all PWH with poor MDR-TB outcomes were simply noncompliant to their medication regimen, especially as South African clinical guidelines include close follow-up and long periods of hospitalization in which medication adherence is tightly controlled. It is also possible that ongoing HIV viral replication could increase inflammation and worsen the immune response to MDR-TB treatment. Future studies should continue to explore and expand upon these potential mechanisms.

South African MDR-TB treatment guidelines recommend HIV viral load testing at MDR-TB diagnosis, after six months of MDR-TB treatment, and then yearly for the duration of MDR-TB treatment if the results remain undetectable, while a detectable viral load requires rapid action including intensive counseling, adherence support, and repeat testing within 2 months [19]. In this study, 33.7% of PWH did not have an available HIV viral load at or within 12 months prior to MDR-TB treatment initiation, and 54.5% did not have an HIV viral load within three months prior to or six months after MDR-TB outcome. One possible explanation for the large proportion of unavailable viral load results could be limited testing due to electronic gatekeeping within the NHLS which limit HIV viral load measurement to yearly by automatically rejecting repeat samples sent within a 12-month window as a cost containment measure. Given the importance of HIV viral suppression in achieving MDR-TB treatment success, measuring HIV viral load routinely and optimizing adherence support and/or ART regimens must become a priority and gatekeeping structures reassessed. Once PWH are identified as having a detectable viral load, clinicians including doctors, nurses, pharmacists, and community health workers must intervene quickly to understand and address factors associated with adherence challenges, intensively counsel patients, and ensure that an effective ART regimen is prescribed. If the viral load is not suppressed following a period of reliable ART adherence, testing for resistance is essential.

There are several limitations to this study. The significant proportion of PWH with unknown HIV viral loads at the time of MDR-TB diagnosis (33.7%) and at the time of MDR-TB outcome (54.5%) limited the power of our study, especially in the second analysis when our sample size was reduced. Though we addressed this issue in the first model by considering a third category of PWH (those with an unknown viral load), a relative risk ratio of MDR-TB outcome calculated from a cohort of PWH in which viral load is known for all would be more precise. Secondly, ART use is known to be the single most important factor predicting HIV viral suppression. In this study, we included all PWH who were diagnosed with MDR-TB regardless of whether they were newly diagnosed or knew their status and whether they were taking ART at the time of MDR-TB diagnosis. These nuances may have

influenced who received an HIV viral load test at the time of MDR-TB diagnosis, as many clinicians in resource-limited settings will not order an HIV viral load for people who are not taking ART, assuming it will be detectable. Thus, ART status may have confounded who received an HIV viral load test at MDR-TB treatment initiation. Given these limitations, we feel that the relative risk ratios for those with a known HIV viral load may be more generalizable to other populations of PWH and MDR-TB, while the results for those with an unknown HIV viral load reinforce the importance of obtaining an accurate viral load measure. Finally, the small number of participants with MDR-TB treatment failure (48, 3.3%) limited the power of our study to determine relationships related to this outcome. While our results show that those with a detectable or unknown HIV viral load at MDR-TB treatment initiation were more likely to experience treatment failure, this relationship did not achieve statistical significance, though it may in a study with a larger number of participants with treatment failure.

## Conclusions

Differences in MDR-TB outcomes for PWH compared to their HIV-negative peers may be driven by HIV viral load status rather than HIV co-infection itself. Those with a detectable HIV viral load at MDR-TB treatment initiation had a significantly increased relative risk of death, and those who either became detectable during MDR-TB treatment or failed to achieve viral suppression by MDR-TB outcome were at increased risk of multiple negative MDR-TB outcomes relative to MDR-TB treatment success, compared to those who achieved or maintained viral suppression. Frequent and timely evaluation of HIV viral load and early intervention, including intensive adherence counseling and optimizing ART regimen, for those with a detectable viral load are essential to reducing the risk of a poor MDR-TB outcome for PWH.

## Supporting information

**S1 Data. All data for 1479 study participants.**
(XLSX)

## Author Contributions

**Conceptualization:** Keri Geiger, Jason E. Farley.

**Data curation:** Amita Patil.

**Formal analysis:** Keri Geiger.

**Funding acquisition:** Jason E. Farley.

**Investigation:** Keri Geiger.

**Methodology:** Keri Geiger, Amita Patil, Chakra Budhathoki, Jason E. Farley.

**Project administration:** Kelly Lowensen.

**Supervision:** Amita Patil, Chakra Budhathoki, Kelly E. Dooley, Kelly Lowensen, Norbert Ndjeka, Jason E. Farley.

**Validation:** Norbert Ndjeka, Jacqueline Ngozo, Jason E. Farley.

**Writing – original draft:** Keri Geiger.

**Writing – review & editing:** Keri Geiger, Amita Patil, Chakra Budhathoki, Kelly E. Dooley, Kelly Lowensen, Norbert Ndjeka, Jacqueline Ngozo, Jason E. Farley.

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
