## [Decision Letter · Decision Letter 0]

16 Nov 2023

PGPH-D-23-01695

Relationship between HIV Viral Suppression and Multidrug Resistant Tuberculosis Treatment Outcomes

Dear Dr. Geiger,

Thank you for submitting your manuscript to PLOS Global Public Health. After careful consideration, we feel that it has merit but does not fully meet PLOS Global Public Health’s publication criteria as it currently stands. Therefore, we invite you to submit a revised version of the manuscript that addresses the points raised during the review process.

We look forward to receiving your revised manuscript.

Kind regards,

Lei Gao

Academic Editor

Journal Requirements:

2. Please ensure you have included the registration number for the clinical trial referenced in the manuscript

Additional Editor Comments (if provided):

Reviewers' comments:

Reviewer's Responses to Questions

**Comments to the Author**

1. Does this manuscript meet PLOS Global Public Health’s publication criteria? Is the manuscript technically sound, and do the data support the conclusions? The manuscript must describe methodologically and ethically rigorous research with conclusions that are appropriately drawn based on the data presented.

Reviewer #1: Yes

Reviewer #2: Yes

2. Has the statistical analysis been performed appropriately and rigorously?

Reviewer #1: Yes

Reviewer #2: Yes

3. Have the authors made all data underlying the findings in their manuscript fully available (please refer to the Data Availability Statement at the start of the manuscript PDF file)?

Reviewer #1: No

Reviewer #2: No

4. Is the manuscript presented in an intelligible fashion and written in standard English?

Reviewer #1: Yes

Reviewer #2: Yes

5. Review Comments to the Author

Reviewer #1: The authors are report on the association between HIV viral load status and outcomes of MDR TB treatment. This is an important topic and as they point out, the effect of HIV on MDR TB outcomes has not previously been described by viral load status. They comment on how missing data was addressed and on the interpretation of effects derived from small sample sizes. Overall an interesting and well-written manuscript.

There are no major comments except to standardise the format of references. Some references list all authors while others are shortenned, and some name authors by initials and surname while others state first names and surnames.

Reviewer #2: Thank you for the opportunity to review this manuscript.

I have minor comments

Lines 69-70, please highlight changes to the standard of care for MDR TB over the time of this project- could this have had an impact on outcomes?

Lines 99. what were the VL changes overtime, ie. as you considered baseline and outcomes VL- what was the picture in the intervening period were fluctuations for some patients and and was this in any way related to outcomes

Lines 149-150- what was the rationale for not inputting VL data while you imputes for some variables

6. PLOS authors have the option to publish the peer review history of their article (what does this mean?). If published, this will include your full peer review and any attached files.

**Do you want your identity to be public for this peer review?** For information about this choice, including consent withdrawal, please see our Privacy Policy.

Reviewer #1: No

Reviewer #2: No

---

## [Decision Letter · Decision Letter 1]

27 Feb 2024

Relationship between HIV Viral Suppression and Multidrug Resistant Tuberculosis Treatment Outcomes

PGPH-D-23-01695R1

Dear Dr. Geiger,

We are pleased to inform you that your manuscript 'Relationship between HIV Viral Suppression and Multidrug Resistant Tuberculosis Treatment Outcomes' has been provisionally accepted for publication in PLOS Global Public Health.

Best regards,

Lei Gao

Academic Editor

Reviewer Comments (if any, and for reference):

Reviewer's Responses to Questions

**Comments to the Author**

1. If the authors have adequately addressed your comments raised in a previous round of review and you feel that this manuscript is now acceptable for publication, you may indicate that here to bypass the “Comments to the Author” section, enter your conflict of interest statement in the “Confidential to Editor” section, and submit your "Accept" recommendation.

Reviewer #1: All comments have been addressed

2. Does this manuscript meet PLOS Global Public Health’s publication criteria? Is the manuscript technically sound, and do the data support the conclusions? The manuscript must describe methodologically and ethically rigorous research with conclusions that are appropriately drawn based on the data presented.

Reviewer #1: Yes

3. Has the statistical analysis been performed appropriately and rigorously?

Reviewer #1: Yes

4. Have the authors made all data underlying the findings in their manuscript fully available (please refer to the Data Availability Statement at the start of the manuscript PDF file)?

Reviewer #1: Yes

5. Is the manuscript presented in an intelligible fashion and written in standard English?

Reviewer #1: Yes

6. Review Comments to the Author

Reviewer #1: No further comments

7. PLOS authors have the option to publish the peer review history of their article (what does this mean?). If published, this will include your full peer review and any attached files.

**Do you want your identity to be public for this peer review?** For information about this choice, including consent withdrawal, please see our Privacy Policy.

Reviewer #1: No
